



# Local and Remote Response of Ozone to Arctic Stratospheric Circulation Extremes

Hao-Jhe Hong[1,2], Thomas Reichler[1]

[1]Department of Atmospheric Sciences, University of Utah, Salt Lake City, 84112, USA
[2]Research Center for Environmental Changes, Academia Sinica, Taipei City, 11529, Taiwan

*Correspondence to*: Hao-Jhe Hong (haojhe.hong@utah.edu)

**Abstract.**

Intense natural circulation variability associated with stratospheric sudden warmings, vortex intensifications, and final warmings is a typical feature of the winter Arctic stratosphere. The attendant changes in transport, mixing, and temperature create pronounced perturbations in stratospheric ozone. Understanding these perturbations is important because of their potential feedbacks with the circulation and because ozone is a key trace gas of the stratosphere. Here, we use MERRA-2 reanalysis to contrast the typical spatiotemporal structure of ozone during sudden warming and vortex intensification events. We examine the changes of ozone in both the Arctic and the Tropics, document the underlying dynamical mechanisms for the observed changes, and analyze the entire life-cycle of the stratospheric events - from the event onset in mid-winter to the final warming in early spring. Over the Arctic and during sudden warmings, ozone undergoes a rapid and long-lasting increase, which only gradually decays to climatology before the final warming. In contrast, vortex intensifications are passive events, associated with decreases in Arctic ozone that gradually intensify during early winter and decay thereafter. The persistent loss of Arctic ozone during vortex intensifications is dramatically compensated by sudden-warming-like increases after the final warming. In the Tropics, the changes in ozone from Arctic circulation events are obscured by the influences from the quasi-biennial oscillation. After controlling for this effect, coherent reductions in tropical ozone can be seen during the onset of sudden warmings, and also during the final warmings that follow vortex intensifications. Our results demonstrate that Arctic circulation extremes have significant local and remote influences on the distribution of stratospheric ozone.

## 1 Introduction

The wintertime Arctic stratosphere is characterized by a number of dynamical, chemical, and physical processes that are coupled to each other in intriguing ways. For example, extreme stratospheric circulation events from the interaction (or lack thereof) of upward propagating planetary-scale Rossby waves with the polar vortex create a pronounced dynamical variability in the Arctic. Large concentration of ozone is another important characteristic of the Arctic stratosphere. Ozone is an effective absorber for solar radiation and an important player in the coupling between the chemistry, radiation, and





dynamics. The diabatic heating from ozone impacts the temperatures and the winds, and the induced dynamical transports
and photochemical reactions again impact the ozone. The feedback between ozone and the circulation may sustain the
circulation anomalies and modify the stratospheric sensitivity to external forcings (Hartmann et al., 2000). Ozone is also
important for the protection of life on Earth by absorbing harmful ultraviolet radiation. Taken together, ozone is a crucial
stratospheric constituent, and understanding the factors that influence its distribution is a critical goal of climate research.

Ozone in the Arctic lower stratosphere is mostly controlled by transports. The transports intensify in the winter
hemisphere (Randel, 1993; Randel et al., 2002) to create a springtime total ozone maximum at high latitudes. The seasonality
of the transports is associated with an intensification of the upward propagating Rossby waves in winter. At times, the bursts
of waves and their interaction with the polar vortex are strong enough to create so-called Stratospheric Sudden Warming
Events (SSWs) (McIntyre, 1982; Limpasuvan et al., 2004; Polvani and Waugh, 2004), arguably the most important form of

stratospheric circulation events. In the process, polar temperatures increase rapidly, reverse the climatological equator-to-
pole temperature gradient, and cause the normal westerly flow of the vortex to become easterly (Scherhag, 1952). SSWs
occur in about two of every three years (Butler et al., 2017), most often in January or February (Horan and Reichler, 2017).

Past studies pointed out the close coupling between the stratospheric dynamics and Arctic ozone (e.g., Leovy et al., 1985;
Ma et al., 2004), with a positive correlation between polar ozone tendencies and the stratospheric wave driving (Randel et

al., 2002). The coupling leads to enhanced poleward ozone transports during SSWs and creates persistent ozone anomalies in
the lowermost stratosphere (Butler et al., 2017; Hocke et al., 2015). De la Cámara et al. (2018) showed that the initial
increase in ozone after SSWs is mainly driven by isentropic eddy fluxes associated with the enhanced wave driving, while
the subsequent recovery of ozone can be attributed to the competing effects between cross-isentropic advection and
irreversible isentropic mixing.

It is perhaps less well-known that the influence of SSWs on ozone can also influence the Tropics. Randel (1993)
demonstrated how vertical transports from the 1979/80 SSW affected tropical ozone in the lower stratosphere and how the
changes in ozone were correlated with temperatures in the upper stratosphere. The SSW-related influences on the Tropics
also imprint on the variability of temperature and water vapor there (Gómez-Escolar et al., 2014; Tao et al., 2015). However,
the SSW effect on tropical ozone is superimposed on the effects from the Quasi-Biennial Oscillation (QBO), downward

propagating westerly and easterly zonal wind anomalies with a cycle of ~28 months (Baldwin et al., 2001; Coy et al., 2016;
Randel and Wu, 1996) that also influence ozone.

The winter Arctic stratosphere not only witnesses occasional SSWs. A sustained lack of stratospheric wave driving can
create the opposite events to SSWs, so-called Vortex Intensification events (VIs). VIs are characterized by an unusually
strong and cold polar vortex (Limpasuvan et al., 2005), and reduced transports of ozone into the pole region (Isaksen et al.,

2012). The extreme cold during VIs favors halogen-induced chemical ozone depletion, which, in combination with the
weakened transport, leads to record low levels of ozone that can be comparable in magnitude to its southern hemispheric
counterpart (Isaksen et al., 2012; Manney et al., 2011). A good example is the most recent winter 2019/20, which





experienced an exceptionally strong, cold, and persistent Arctic stratospheric polar vortex, and which led to record-breaking Arctic ozone depletion.

Another important class of stratospheric circulation events are stratospheric Final Warming events (FWs). FWs occur every year at the end of winter, representing the final breakdown of the polar vortex due to the seasonal increase in solar heating. FWs are often triggered by pulses of increased wave activity and can be considered as SSWs that conclude the winter season (Black et al., 2007). There also exists an interesting temporal relationship between FWs, SSWs, and VIs: FWs that are preceded by SSWs in the same winter tend to occur significantly later than the mean FW date (~mid-April, Horan

and Reichler, 2017), and FWs that are preceded by non-SSW winters (i.e., neutral winter and VIs) tend to be relatively early (Hu et al., 2014). This can be explained from the delayed relationship between vortex strength and wave driving. An SSW, for example, is usually followed by reduced wave activity and hence a stronger vortex, which then breaks down later in spring. The changes in FW timing also impact the levels of Arctic ozone: Manney and Lawrence (2016) showed that the chemical ozone loss from the 2016 VI was disrupted by an early FW at the beginning of March and suggested that FWs may

have comparable effects on Arctic ozone as SSWs.

    While the aforementioned studies have started to investigate the response of ozone in the Arctic to SSWs, the response of ozone in the Tropics and also to VI and FW events has received little attention so far. This study intends to fill this gap and refine the existing knowledge about the spatiotemporal relationship between ozone and a range of Arctic stratospheric circulation events using a modern observation-based perspective. We achieve this by taking a comparative approach that

contrasts the often-opposing ozone behavior between SSWs and VIs, and between the Arctic and the Tropics. Time is another distinctive aspect of this study, as we cover the entire life-cycle of the stratospheric circulation events from the event onset in the middle of winter to the date of the FW at the end of winter. We also clarify the role of the associated dynamical and photochemical processes in changing ozone. Overall, our goal is to provide an up-to-date observation-based view of the global natural dynamics-driven variability of stratospheric ozone. This is not only of interest in its own right but also

provides an observational baseline for ozone behavior during stratospheric circulation events that can be used for the validation of coupled chemistry-climate models.

    This paper is structured as follows. In Sect. 2, we describe the data and methods used in this study. In Sect. 3, we demonstrate the ozone response in the Arctic, while in Sect. 4 we continue our discussion for the Tropics. A summary and conclusion are provided in Sect. 5.

## 2 Data and Methods

### 2.1 MERRA-2 Data

We use 1980-2018 daily fields from the MERRA-2 reanalysis (Bosilovich et al., 2015) at a horizontal resolution of 1.5º and 37 levels ranging from 1000 to 0.1 hPa. MERRA-2 also provides ozone, which is based on retrievals from the SBUV (January 1980-September 2004) and Aura MLS/OMI (October 2004-present) instruments (Davis et al., 2017) and on a





simple ozone scheme (Rienecker et al., 2008). MERRA-2 has been shown to perform well for ozone through much of the stratosphere (Davis et al., 2017; Wargan et al., 2017). Most of our calculations are based on zonal mean quantities. We compute daily climatologies from MERRA-2 by averaging each day of the year over the entire record and smoothing over the seasonal cycle using 10-day running means. Daily anomalies are obtained by subtracting the climatologies from the daily data.

## 2.2 Event Definition

In defining SSWs and FWs, we follow the widely used prescription by Charlton and Polvani (2007). An SSW is detected when the zonal-mean zonal wind at 10 hPa and 60° N (U1060) switches from westerly to easterly (the central date of the event) during November-March and returns to westerly for at least ten consecutive days before 30 April. If the return to westerly condition is not fulfilled, the event is considered as the FW of the year. Two or more SSWs in the same winter must
be separated by consecutive westerlies for at least 20 days. Since we are interested in the evolution of ozone over the life-cycle of SSWs from the middle to the end of the winter, we only consider mid-winter SSWs during January or February. We also discard mid-winter SSW events that are followed by another, potentially disturbing, SSW, leading to the exclusion of only one event.

Our definition of mid-winter VIs is also based on U1060, but we first low-pass filter the data using 20-day running means.
A mid-winter VI occurs when the smoothed daily U1060 anomaly during January or February exceeds one standard deviation (16 m s-1), marking the central date of the VI. Like SSWs, two VIs in the same winter must be separated by at least 20 days. We only consider VIs that are not followed by another VI or SSW.

As shown in Table 1, our definitions lead to 15 SSWs and 8 VIs. For SSWs, the mean central date and the associated FW date are 3 February and 26 April, respectively, leading to a mean duration of 83 days (ranging from 54 to 117). VIs have a
mean central date on 23 January and an associated FW date on 2 April. This translates into a mean duration of 70 days (ranging from 44 to 91). Note that SSWs are longer in duration than VIs, consistent with the findings by Hu et al. (2014) that SSW winters are associated with FW dates that are on average late compared to the climatological mean FW date.

We use the 180-day smoothed zonal-mean equatorial (±5°) zonal wind at 30 hPa (UEQ30) to determine the phase of the QBO. A QBO cycle is defined as the period between two consecutive positive UEQ30 maxima, and the UEQ30 minimum in
between is considered as the midpoint of the cycle. We exclude the anomalous QBO cycle of 2015-2016 (Newman et al., 2016) from our analysis and obtain 16 QBO cycles over the 1980-2018 period.

## 2.3 Ozone and Dynamics Diagnostics

The changes in zonal-mean ozone ($\bar{\chi}$) are investigated using the transformed Eulerian mean (TEM) approach. Following Andrews et al. (1987), the TEM tracer transport equation in p-coordinates
$$\bar{\chi}_t = -\bar{v}^* \bar{\chi}_y - \bar{\omega}^* \bar{\chi}_p - \rho_0^{-1} \boldsymbol{\nabla} \cdot \boldsymbol{M} + \bar{S}, \tag{1}$$





is used to decompose the ozone tendency ($\bar{\chi}_t$) into two advection terms associated with the residual mean circulation, one term due to eddy flux convergence ($-\rho_0{}^{-1}\nabla \cdot \boldsymbol{M}$), and a source term ($\bar{S}$) that represents the effects of chemistry on ozone. The eddy flux convergence term represents the effects of the resolved eddies in transporting and mixing ozone (Andrews et al., 1987). Here, $\bar{v}^*$ and $\bar{\omega}^*$ are the components of the residual mean circulation, $\rho_0$ is the basic state density, $\boldsymbol{M}$ is an eddy

flux vector given by

$$\boldsymbol{M} = [\rho_0(\overline{v'\chi'} - \overline{v'\theta'}\,\bar{\chi}_p/\bar{\theta}_p), \rho_0(\overline{\omega'\chi'} + \overline{v'\theta'}\,\bar{\chi}_y/\bar{\theta}_p)]\,,$$

where overbars denote zonal means, primes are deviations from zonal means, and the other terms are standard notation.

The ozone tendency $\bar{\chi}_t$ is calculated by taking forward differences in time of daily ozone, and the chemical source term $\bar{S}$ is the residual between $\bar{\chi}_t$ and the sum of the three dynamical terms of Eq. (1). We note that the resulting $\bar{S}$ does not

exclusively reflect the chemical production or destruction of ozone because of unavoidable errors of MERRA-2 and computational uncertainties. For example, in the absence of observations, the MERRA-2 ozone is calculated from a simple parameterization (Rienecker et al., 2008), which can result in considerable errors. Because of this uncertainty, and also because of the focus of this study on the dynamical impacts, we do not show the $\bar{S}$ term.

We use $F_p$, the vertical component of the quasi-geostrophic Eliassen-Palm (EP) flux (Eliassen and Palm, 1961), to

diagnose the upward propagating Rossby wave activity. Following Andrews et al. (1987), $F_p$ is given by

$$F_p = -a\cos\phi f \frac{\overline{v'\theta'}}{\theta_p}\,, \tag{2}$$

where all symbols are standard notation. In our analysis, we reverse the sign of $F_p$ so that positive $F_p$ corresponds to upward propagation. We focus on $F_p$ at 100 hPa averaged over 40° N-80° N and refer to this quantity as the stratospheric wave driving.

**2.4 Event Compositing**

Traditional composites take averages of various events centered on specific dates (e.g., Butler et al., 2017). However, in the present study, we are interested in the behavior of ozone during the entire life cycle of stratospheric circulation events, with each event having a somewhat different duration. The life cycle starts at the central date and ends with the FW at the end of winter. Since the duration (i.e., the time between the central date and the FW) differs from event to event, we somewhat

modify the traditional compositing technique. Our approach is based on the mean central date of all selected SSWs (or VIs) ($\bar{t}_0$) and the mean date of their associated FWs ($\overline{t_{FW}}$). We then use linear interpolation in time to align the dates of the individual events ($t_0, t_{FW}$) with the composite mean dates ($\bar{t}_0, \overline{t_{FW}}$). Mathematically, this can be written as

$$\bar{t} = \bar{t}_0 + (t - t_0) * \frac{\overline{t_{FW}} - \bar{t}_0}{t_{FW} - t_0}\,, \tag{3}$$

where $\bar{t}$ denotes the time of the composite and $t$ the time of individual events. The interpolation can be interpreted as a

stretching or squishing of the time axis so that all data during $t_0$ ($t_{FW}$) are aligned with $\bar{t}_0$ ($\overline{t_{FW}}$). The mean duration ($\overline{t_{FW}} - \bar{t}_0$) of SSWs (VIs) is then 83 (70) days. The duration of individual events ($t_{FW} - t_0$) is shown in Table 1. We use this



technique to create composites of various quantities at daily intervals. A one-tailed Student's t-test at the 95% confidence level is used to test the statistical significance of the composite mean anomalies against the null hypothesis of zero anomalies.

**3 Arctic Ozone**

**3.1 Arctic Circulation Changes**

We begin our discussion of how Arctic ozone evolves during SSWs and VIs by presenting some key dynamical quantities, which will then guide the interpretation of our subsequent results. Fig. 1 shows the evolution of composite anomalies in the stratospheric wave driving (top), the vertical component of the residual circulation (middle), and temperature (bottom) over

the life cycle of SSWs (left) and VIs (right).

SSWs (Fig. 1, left) are typically preceded by enhanced stratospheric wave driving, starting at a negative lag of ~15 days (Fig. 1a). This leads to the breakdown of the polar vortex and marks the onset of the SSW (Limpasuvan et al., 2004). After the onset, the wave driving decreases rapidly and becomes negative, contributing to the over-recovery of the vortex in the upper stratosphere, reminiscent of so-called polar-night jet events (Kuroda and Kodera, 2001; Hitchcock and Shepherd,

2013). As pointed out by Plumb and Eluszkiewicz (1999) and demonstrated by Fig. 1c, this cyclic nature of the wave driving imprints on the residual circulation of the entire stratosphere. Fig. 1c shows that the vertical component of the residual circulation ($\bar{\omega}^*$) over the Arctic varies consistently with the wave driving, with enhanced downwelling during onset (reddish colors), followed by a long period of enhanced upwelling (bluish colors). The cycle ends at the end of winter, with somewhat enhanced wave driving and subsequent downwelling during the FW. Arctic temperatures (Fig. 1e) are characterized by

cooling before the SSW, strong warming in the middle to lower stratosphere during and after the onset, and cooling after the onset in the upper to the middle stratosphere. The patterns of warming and cooling following the onset give the impression of a downward propagation. However, the cooling in the upper stratosphere is associated with the aforementioned suppressed wave driving and subsequent radiative cooling (Hitchcock and Shepherd, 2013; Limpasuvan et al., 2004), and the persistence of the warming in the lower stratosphere is related to the long radiative time scale in this part of the stratosphere.

VIs (Fig. 1, right) are in many respects opposite to SSWs. As explained in Limpasuvan et al. (2005), VIs evolve relatively slowly and result from the sustained lack of stratospheric wave driving, leading to the gradual strengthening and cooling of the vortex. As shown by Fig. 1b, the wave driving is anomalously small, starting several weeks before onset and minimizing at about one week after onset. This is different from SSWs, as the wave driving during SSWs changes much more abruptly during onset. Long after the onset of VIs, the wave driving increases again, first more intermittently, and then more

systematically during the FW. We note that the magnitude of the wave driving associated with the FW is quite large and comparable to that of SSWs during onset. This may be attributable to the sustained suppression of wave driving during VI onset, contributing to the enhanced release of wave activity after the event and a relatively early FW. Also, the relatively strong polar vortex after VIs (not shown) is conducive for upward propagating wave activity into the stratosphere.



As for SSWs, changes in the Arctic $\overline{\omega}^*$ during VIs (Fig. 1d) agree well with the evolution of wave driving. The upwelling
maximizes one week after VI onset, followed by a period of intermittent downwelling before the FW (see also Limpasuvan et al., 2005). VIs are also associated with pronounced and persistent Arctic cooling (Fig. 1f) in the lower stratosphere, which is in contrast to the significant warming that starts about one week before VI onset in the upper stratosphere. The warming slowly propagates downward, persists until spring, and finally becomes part of the FW that concludes the winter season. The timing and strength of the FW is another important difference between SSWs and VIs. While FWs after SSWs tend to be late
and mostly represent a transition into climatology, FWs after VIs occur early, are relatively strong, and contribute to a pronounced weakening and warming of the vortex.

### 3.2 Arctic Ozone Changes

The above-described dynamical perturbations are associated with significant changes in transport of stratospheric ozone and its temperature-dependent photochemical reaction rates. As has been shown to some extent before (Butler et al., 2017; de la
Cámara et al., 2018; Hocke et al., 2015), and as we will show in more detail next, this has major consequences for the distribution of stratospheric ozone.

We first examine the composite evolution of Arctic column ozone (i.e., the vertically integrated ozone amount) during SSWs (Fig. 2a). Red and gray shading indicate the deviation of the column ozone from its climatology (thick black curve), and the green line shows the anomalous column ozone tendency. As noted by Randel et al. (2002), the ozone tendency is
well correlated with the wave driving (Fig. 1a). Before onset, there is a subtle decrease in column ozone, presumably related to the anomalously strong and cold vortex during this time (Fig. 1e) and the reduced ozone transport into the polar regions. Within the first 10 days following the SSW onset, the column ozone anomalies rapidly increase by ~50 DU and persist for up to 60 days until late winter. Hocke et al. (2015) suggested that the increases in column ozone after SSWs amount to up to 90 DU over the Arctic, which is nearly twice of what we find. The vertically resolved Arctic ozone mixing ratio (Fig. 2c)
shows a more complicated picture. There is a pronounced reduction in ozone in the middle and upper stratosphere after SSWs, which seems to be slowly descending downward. This decrease in mid-stratospheric ozone, which starts about one month after SSWs, has also been noted by Sagi et al. (2017). Ozone in the upper stratosphere also undergoes a complicated evolution. The negative anomalies above 5 hPa exist only shortly during onset. They are followed by persistent positive anomalies, which again tend to descend downward by mid-March, diminish by April, and reemerge at mid-stratospheric
levels by the end of April as a consequence of the FW.

Next, we examine the evolution of Arctic ozone during VIs (Fig. 2, right). Column ozone (Fig. 2b) is anomalously negative over the entire VI life cycle, minimizing at about -40 DU by mid-March. As with SSWs, the anomalous tendency of column ozone (green line) resembles the wave driving (Fig. 1b) and is strongly positive during the FW. Fig. 2d demonstrates that the negative ozone anomalies maximize in the middle stratosphere at ~10 days after onset and also tend to propagate
downward into the lower stratosphere. These anomalies are particularly long-lasting in the lower stratosphere, where they exist for more than 60 days until the FW. This composite behavior is very similar to the case study by Manney and Lawrence



(2016), who reported that the rapid Arctic chemical ozone loss during winter 2015/16 was abruptly terminated by the early FW in March. Ozone anomalies are also negative in the upper stratosphere, where they persist throughout the VI life-cycle and tend to descend after the FW. At the FW, there are strongly positive ozone anomalies in the middle stratosphere. The

structure of these anomalies is similar to that of SSWs, except that the anomalies are weaker and do not appear in the lower stratosphere.

We now explore the role of the dynamical mechanisms that create the changes in ozone. From the TEM tracer transport equation Eq. (1) it is clear that several processes are involved. Fig. 2e-2j present the total time tendencies of ozone (e-f) and the contributions to it from vertical advection (g-h) and eddy flux convergence (i-j). The horizontal advection term is

generally small and therefore omitted. For better orientation, the red and blue contours reproduce a constant ozone mixing ratio anomaly from Figs. 2c and 2d.

The negative Arctic ozone anomalies in early winter before SSWs are partly the result of reduced eddy flux convergences (Fig. 2i) and vertical transports (Fig. 2g). The strong positive ozone tendencies close to the onset of SSWs, which are responsible for the increase in ozone after SSWs, result mainly from the convergence of eddy fluxes (Fig. 2i) (see also de la

Cámara et al., 2018), triggered by the enhanced wave driving associated with SSWs (Fig. 1a). The downward transport of ozone by the enhanced residual circulation also contributes to the positive tendencies during onset, in particular in the lower stratosphere (Fig. 2g). After SSWs, the suppressed planetary wave activity leads to a sustained reduction of eddy transports, and hence negative ozone tendencies in the middle and lower stratosphere. At the same time, the vertical advection of ozone is anomalously negative in the middle stratosphere after SSWs. Both effects lead to the gradual decay of the strongly positive

ozone anomalies right after onset and eventually create the abovementioned banded structure of negative ozone in the middle stratosphere. This indicates that the decrease in mid-stratospheric ozone after SSWs is mainly of dynamical origin, whereas Sagi et al. (2017) argue that it is due to chemical reactions involving NOx species. During the time of the FW, the eddy flux convergence becomes somewhat positive (Fig. 2i), overall leading to ozone mixing ratios that are close to climatology. In the upper stratosphere, the temperature-dependent photochemistry plays a dominant role for ozone. There, ozone is mostly anti-

correlated with temperature (Craig and Ohring, 1958), which can be seen by comparing Fig. 1e (for temperature) with Fig. 2c (for ozone).

The VI related total Arctic ozone tendencies (Fig. 2f) are mostly equal but opposite in sign to that of SSWs. VIs are passive events that develop gradually by radiative cooling out to space, and the related negative ozone anomalies appear long before the actual onset (Fig. 2d), related to periods of negative tendencies before and during VI onset (Fig. 2f). The

tendencies are related to reduced eddy transports in the upper half (Fig. 2j) and reduced vertical advection in the lower half of the stratosphere (Fig. 2h). Ozone in the upper stratosphere slowly recovers towards climatology, mostly due to increases in eddy transport associated with pulses of planetary waves that restore the vortex back to normal. However, the positive eddy transport is counteracted by the photochemical effect as the temperature is anomalously warm in this layer (Fig. 1f). In contrast, the negative ozone anomalies in the lower stratosphere are sustained by reduced vertical advection (Fig. 2h) until

mid-March. As explained before, FWs that follow VIs tend to be relatively strong and somewhat resemble SSWs, leading to





sizeable increases in Arctic ozone. As with SSWs, this is associated with positive eddy transports in the upper half (Fig. 2j) and positive vertical advection in the lower half of the stratosphere (Fig. 2h). The two effects compensate for the prior ozone deficits, leading to an overall recovery of the column ozone anomalies (Fig. 2b).

## 4 Tropical Ozone

### 4.1 Tropical Circulation Changes

We now turn our attention to the Tropics, defined as the ±15° latitude band. Tropical ozone is changing in response to Arctic circulation events because of the global nature of the meridional overturning and its role in the transport of ozone (Randel, 1993). We start our discussion by focusing on the changing dynamics in the Tropics during Arctic circulation events (Fig. 3). During SSWs, the variations of $\overline{\omega}^*$ (Fig. 3a) are largely opposite to that in the Arctic (Fig. 1c), except during the time of the FW. This demonstrates that the global nature of the enhanced residual circulation during SSWs also affects the Tropics, leading to stronger upwelling and cooling. The cooling persists in the lower stratosphere, but quickly transitions into warming in the middle and upper stratosphere (Fig. 3c) (see also Gómez-Escolar et al., 2014; Tao et al., 2015).

In comparison with the SSWs, the variations of $\overline{\omega}^*$ during VI onset (Fig. 3b) are less well synchronized with that in the Arctic (Fig. 1d), perhaps due to the relative weakness of the wave driving and also due to influences from the QBO. Although $\overline{\omega}^*$ is quite noisy, temperatures during VI onset show significant warming in the tropical lower stratosphere (Fig. 3d), probably related to adiabatic warming from anomalous downwelling (Fig. 3b). By mid-February, a downward propagating cooling anomaly can be seen in the tropical upper stratosphere (Fig. 3d), as one would expect from the anomalous upwelling (Fig. 3b). As noted before, FWs after VIs are dynamically similar to SSWs, and this is also noticeable in the Tropics. For example, the enhanced extratropical wave driving at the FW is also reflected in the tropical $\overline{\omega}^*$.

### 4.2 QBO Influences on Tropical Ozone

Understanding the changes in tropical ozone in response to Arctic stratospheric circulation events is complicated by the simultaneous influences from the QBO. To disentangle the two effects, we first examine how the vertical structure of tropical ozone changes in response to the QBO. Fig. 4a shows the vertical cross-section of tropical ozone anomalies (±15°) composited on the phase of the QBO from 16 QBO cycles. The black curve represents the mean evolution of UEQ30, where a QBO cycle is defined by two consecutive maxima in UEQ30. Assuming a mean QBO period of 28 months (Baldwin et al., 2001), a one-degree phase change of the QBO corresponds to ~2.3 days. Tweedy et al. (2017) performed a similar analysis (their Fig. 1) by defining the central month of a QBO cycle from changes in the vertical wind shear at 40 hPa and taking QBO composites for different lags. Our results (Fig. 4a) are in good agreement with their study, e.g., positive zonal winds at 30 hPa are associated with increasing ozone in the lower stratosphere. Our result also agrees with Baldwin et al. (2001), that maximum column ozone values occur when the westerly wind shear descends into the lowermost stratosphere. The vertical


structure of the QBO ozone anomalies in Fig. 4a also shows two maxima at ~10 hPa and ~30 hPa, shifted by about a quarter QBO cycle, consistent with previous findings (Coy et al., 2016; Randel and Wu, 1996).

Fig. 4b demonstrates that SSWs and VIs occur during virtually any phase of the QBO, making it difficult to cleanly separate the ozone changes from the Arctic and the QBO. However, as shown by the mean timing of the events (V and S markers at the right), there is a slight preference for SSWs to occur during the easterly QBO phase and VIs during the westerly QBO phase, a possibility that was discussed by Dunkerton et al. (1988). We tested various approaches for filtering out the QBO influences from the tropical ozone. In one approach, we subtracted the composite QBO ozone signal (Fig. 4a) from the daily tropical ozone anomalies during SSWs or VIs according to the QBO phase of each event. But the outcomes were unsatisfying, probably because of large event-to-event variability of the QBO ozone. We also tried to follow Kodera (2006) and subtracted the 61-day mean ozone anomaly centered on the onset date of each circulation event, but this also reduced much of the ozone signals associated with the events. Finally, we tested a method proposed by Gómez-Escolar et al. (2014) and subtracted the preexisting ozone anomalies of each event from its subsequent daily ozone fields. We found by experimenting that using days -60 to -30 with respect to the central dates leads to the best results, and we used this time interval for preparing Figs. 5c and 5d.

## 4.3 Tropical Ozone Changes

Fig. 5 presents composite anomalies and composite anomalous tendencies in tropical ozone during SSWs and VIs. For SSWs, the column ozone tendency (Fig. 5a, green line) is as expected anti-correlated with the extratropical EP flux (Fig. 1a), suggesting a direct influence of SSWs on tropical ozone through enhanced residual circulation. SSWs are followed by a reduction in tropical column ozone by ~2.5 DU and an increase by ~1-2 DU after mid-March, which persists until late spring. Fig. 5c shows the vertically resolved composite for tropical ozone after removing the preexisting ozone signal from the QBO, indicating that the local tropical ozone anomalies associated with SSWs are confined to levels above ~60 hPa. During SSW onset, the response of ozone is characterized by significant increases in the upper stratosphere and decreases below the middle stratosphere (~10 hPa), roughly opposite to that in the Arctic (Fig. 2c). The ozone anomalies reverse sign after mid-February and persist into late spring.

During VIs (Fig. 5b), the tropical column ozone anomalies are mostly positive (~1 DU) and only become negative (~2 DU) after the FW. However, the vertically resolved ozone anomalies with the QBO influence removed (Fig. 5d) show a weak dipole in the middle stratosphere around the onset, with little response in the lower stratosphere. This indicates that the increased column ozone anomalies in Fig. 5b are likely due to the QBO. As discussed before, the weak tropical ozone response to VIs is linked to the relative weakness of the wave driving during VIs, which is not sufficient to affect the tropical upwelling. However, during the FW of VIs, the wave driving anomaly is strong enough; the resulting tropical ozone response is similar to that during SSW onset, with a strong and persistent dipole centered at ~20 hPa.

The dynamical mechanisms that create the changes in tropical ozone are dominated by vertical advection associated with changes to the residual circulation (Randel, 1993). Enhanced tropical upwelling during SSW onset (Fig. 3a) combined with a





vertical background of ozone mixing ratios that maximize in the middle stratosphere create positive tendencies above 10 hPa
and negative tendencies below 10 hPa (Fig. 5g). Following the reversal of the residual circulation anomalies at about 10 days
after onset (Fig. 3a), the vertical advection term leads to oppositely-signed ozone anomalies starting at about mid-February.
During VIs, the tropical ozone tendencies (Fig. 5f) are mostly small. There are negative tendencies from vertical advection
(Fig. 5h) in the upper stratosphere and during onset, owing to the weakened meridional circulation from the VI. However,
these negative tendencies are compensated by the chemical source term (not shown), overall leading to little changes in
ozone. As expected, the tropical ozone tendencies during the FW of VIs (Fig. 5f) are mostly due to vertical advection (Fig.
5h) and compensating influences from the source term $\bar{S}$ (not shown).

## 5 Summary and Conclusion

We used MERRA-2 reanalysis to document the composite spatiotemporal ozone response to Arctic circulation events. While
the ozone response in the Arctic to Sudden Stratospheric Warming (SSW) events has already been the target of some
previous studies (Butler et al., 2017; de la Cámara et al., 2018; Hocke et al., 2015), we took a more holistic approach and
studied stratospheric ozone in the Arctic and the Tropics, and we considered not only SSWs but also Vortex Intensification
(VI) and Final Warming (FW) events.

In the Arctic, the onset of SSWs leads to a rapid increase of total ozone by ~50 DU, which over the course of ~60 days
gradually transitions towards climatology before the subsequent FWs. Diagnostic analysis using the TEM tracer transport
equation indicates that through the entire life cycle of SSWs, ozone transports by eddies prevail over vertical transports from
the anomalous mean meridional circulation. In contrast, during VIs, Arctic ozone exhibits a slow but progressive decrease,
which begins in early winter and results in a ~40 DU reduction by mid-March. The strongest negative ozone tendencies take
place right after the central date of VIs, attributable to weakened vertical transports in the lower stratosphere and decreased
eddy transports in the upper stratosphere. VIs conclude the winter with a relatively early and strong FW, resembling a mid-
winter SSW in terms of the dynamics and ozone perturbations. In contrast, FWs that follow SSWs are relatively late and less
spectacular, representing mostly a smooth transition according to climatology. SSWs have also distinct ozone impacts in the
Tropics. By removing signals attributable to the QBO, we found tropical ozone responses to SSWs that are largely
concurrent and inverse to their Arctic counterparts. At SSW onset, tropical ozone decreases below 10 hPa and increases
above, with an opposite behavior after ~20 days when the residual circulation reverses and persists toward the FW. VIs show
some obscure tropical ozone responses during onset, presumably due to the relatively weak planetary wave driving
anomalies. However, during the FW, VIs are associated with pronounced tropical ozone anomalies due to enhanced vertical
transports.

There are also some limitations to this study. In terms of the mechanisms, we were mostly focused on the various
dynamical effects in changing ozone. However, chemical effects are likely to play also some role in perturbing ozone, in
particular in the chemically-dominated upper stratosphere. We were unable to investigate the chemical effects because of the




large uncertainties associated with the chemical term in the MERRA-2 reanalysis, but we suspect that the dynamics are overall more important than the chemistry. This is supported by Isaksen et al. (2012), who found that the chemical effect explained only 23% of the Arctic ozone loss during the VI from 2011. Nevertheless, it would be interesting to evaluate the relative contributions from the dynamics and the chemistry in changing ozone using output from coupled chemistry climate
models (CCMs). We also did not explicitly consider so-called Downward planetary Wave Coupling events (DWCs) (Lubis et al., 2017) in our study. However, we consider DWCs as less effective in influencing ozone. DWCs are relatively short-lived (< 10 days), associated with increases in ozone before and decreases during the event, leading to a relatively small net response.

One of the novel results of this study is that FWs that follow VIs induce a surprisingly strong ozone response, which
resembles in many respects that of mid-winter SSWs. Another relatively new aspect of this study is that Arctic circulation events also perturb ozone in the Tropics, which is most pronounced during SSWs and early FWs after VIs. This adds to an increasing body of evidence that the mean meridional circulation communicates the effects of Arctic stratospheric circulation events into the lower latitudes. This leads to the notion that the Arctic circulation extremes have an almost global reach, as also evidenced by their impacts on equatorial stratospheric temperatures (Dhaka et al., 2015) and tropospheric equatorial
convective activity (Kodera, 2006). It still remains to be seen how the tropical circulation is affected by the combined heating effects from the tropical ozone and the meridional circulation.

Recent studies have suggested that the dynamical coupling between the stratosphere and the troposphere and the surface impact of this coupling is simulated more strongly in models with interactive ozone chemistry (i.e., CCMs) (Haase and Matthes, 2019; Li et al., 2016; Romanowsky et al., 2019), suggesting that intraseasonal variations of ozone are important for
the prediction of short-term climate. The results from our study could serve as a reference for the validation of CCMs. Simulations with CCMs in turn could be used to clarify some of the still open questions of the present study, in particular about the response of tropical ozone during VIs and the relative role of photochemistry in changing ozone during the circulation events.

**Data availability**

MERRA-2 reanalysis are available online via NASA's Goddard Earth Sciences Data and Information Services Center archive (https://gmao.gsfc.nasa.gov/reanalysis/MERRA-2/data_access/, Bosilovich et al., 2015).

**Author contribution**

HH performed the analysis and wrote the manuscript. TR designed the study, provided guidance in the interpretation of the results, and reviewed the manuscript.



**Competing interests**

The authors declare that they have no conflict of interest.

**Acknowledgements**

We thank the Department of Atmospheric Sciences at the University of Utah for its support. The use of compute infrastructure from the Center for High Performance Computing at the University of Utah is gratefully acknowledged. We
also acknowledge NASA for providing the MERRA-2 reanalysis.

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





**Table 1. Central dates $t_0$ of SSWs and VIs. Numbers in parentheses indicate event duration (in days) between the central date and the following FW, i.e., $t_{FW}$-$t_0$.**

| No. | SSW Central Date | VI Central Date |
| --- | --- | --- |
| 1 | 24 Feb 1984 (61) | 3 Jan 1983 (88) |
| 2 | 1 Jan 1985 (82) | 19 Jan 1993 (83) |
| 3 | 23 Jan 1987 (99) | 17 Feb 1994 (44) |
| 4 | 21 Feb 1989 (54) | 29 Jan 1996 (72) |
| 5 | 26 Feb 1999 (66) | 29 Jan 1997 (91) |
| 6 | 11 Feb 2001 (88) | 9 Jan 2005 (62) |
| 7 | 17 Feb 2002 (77) | 7 Feb 2011 (57) |
| 8 | 18 Jan 2003 (86) | 6 Jan 2016 (59) |
| 9 | 5 Jan 2004 (116) | |
| 10 | 21 Jan 2006 (106) | |
| 11 | 24 Feb 2007 (54) | |
| 12 | 22 Feb 2008 (69) | |
| 13 | 24 Jan 2009 (106) | |
| 14 | 6 Jan 2013 (117) | |
| 15 | 12 Feb 2018 (63) | |
| Mean | 3 Feb (83) | 23 Jan (70) |



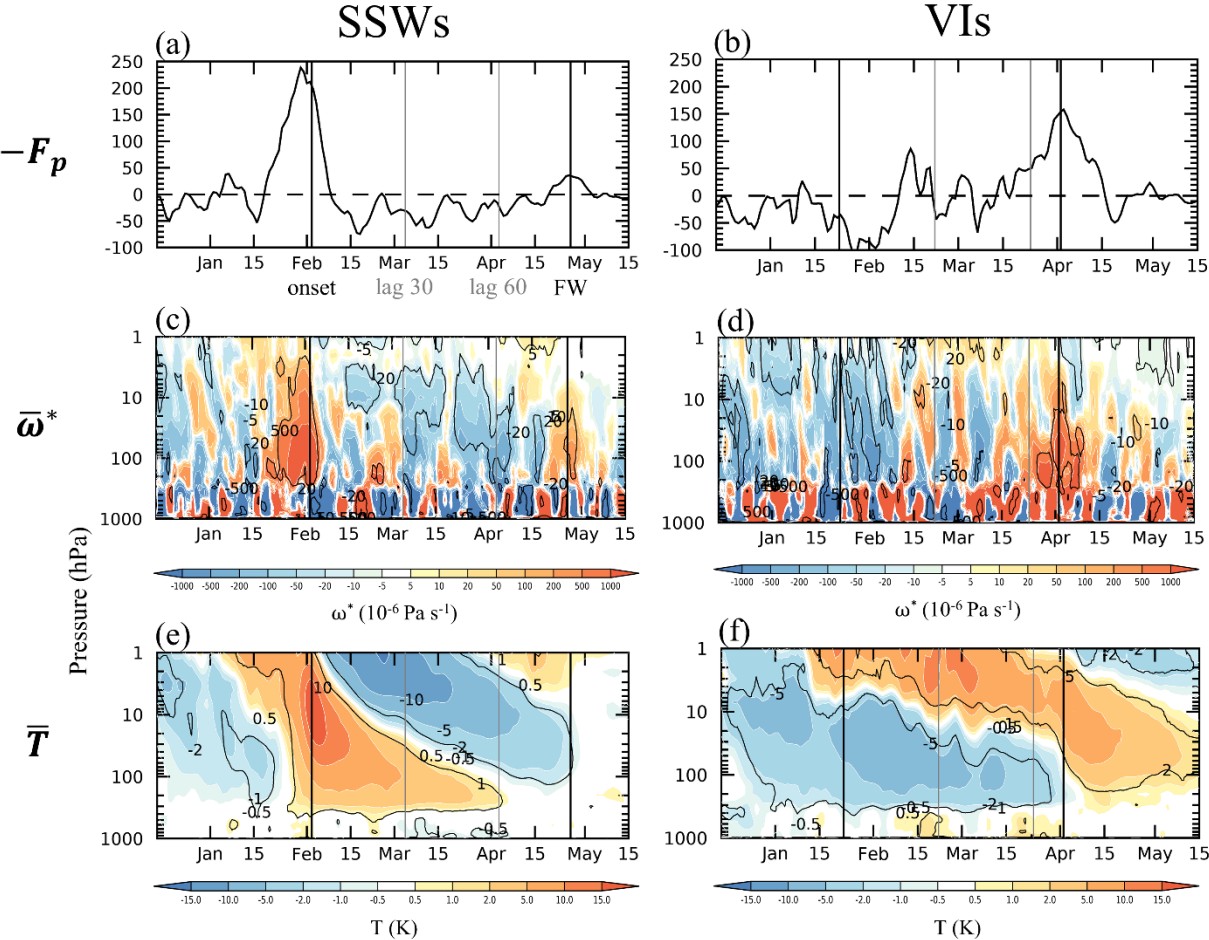

**Figure 1. SSW (left) and VI (right) composites over the Arctic. Shown are (a-b) time series of 10-day smoothed vertical EP flux (10⁴ kg m s⁴) averaged over 40° N-80° N at 100 hPa, and time-height cross-sections for (c-d) vertical component of the residual circulation (10⁻⁶ Pa s⁻¹) (65° N-85° N) and (e-f) temperature (K) (65° N-90° N). Contours represent statistical significance at the 95% level.**





**Figure 2.** Arctic ozone composites during (left) SSWs and (right) VIs. (a-b) Column ozone (left axis) and associated anomalous tendency (right axis); the horizontal line is zero tendencies. Remaining panels are time-height cross-sections of (c-d) ozone mixing ratio ($10^{-2}$ ppmv), (e-f) overall ozone tendency, ozone tendency due to (g-h) vertical advection and (i-j) eddy flux convergence (ppbv day$^{-1}$). Quantities are averaged over 65° N-90° N for ozone and 65° N-85° N for tendencies. Contours in (c-d) represent statistical significance at the 95% level and in (e-j) the ±0.1 ppmv ozone anomalies from (c-d).





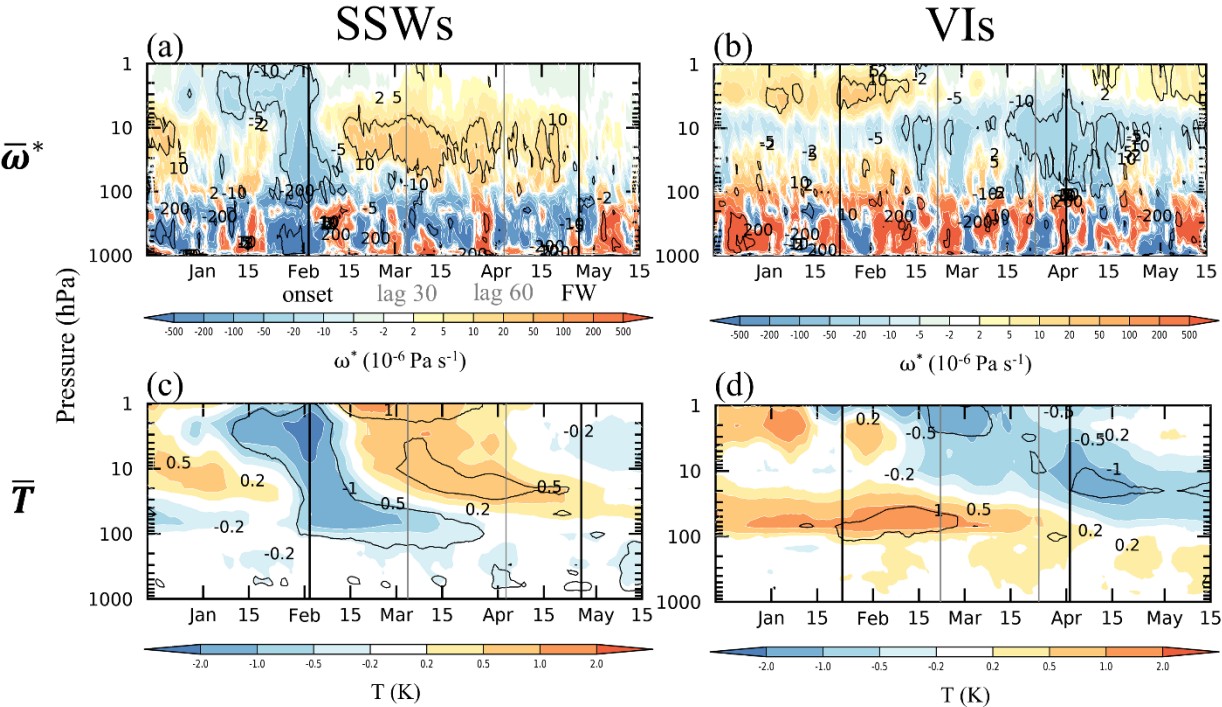

**Figure 3. Composite anomalies for (left) SSWs and (right) VIs over the tropical belt (±15°). Shown are time-height cross-sections for (a-b) the vertical component of the residual mean circulation (10⁻⁶ Pa s⁻¹) and (c-d) temperature (K). Contours are as in Fig. 1.**





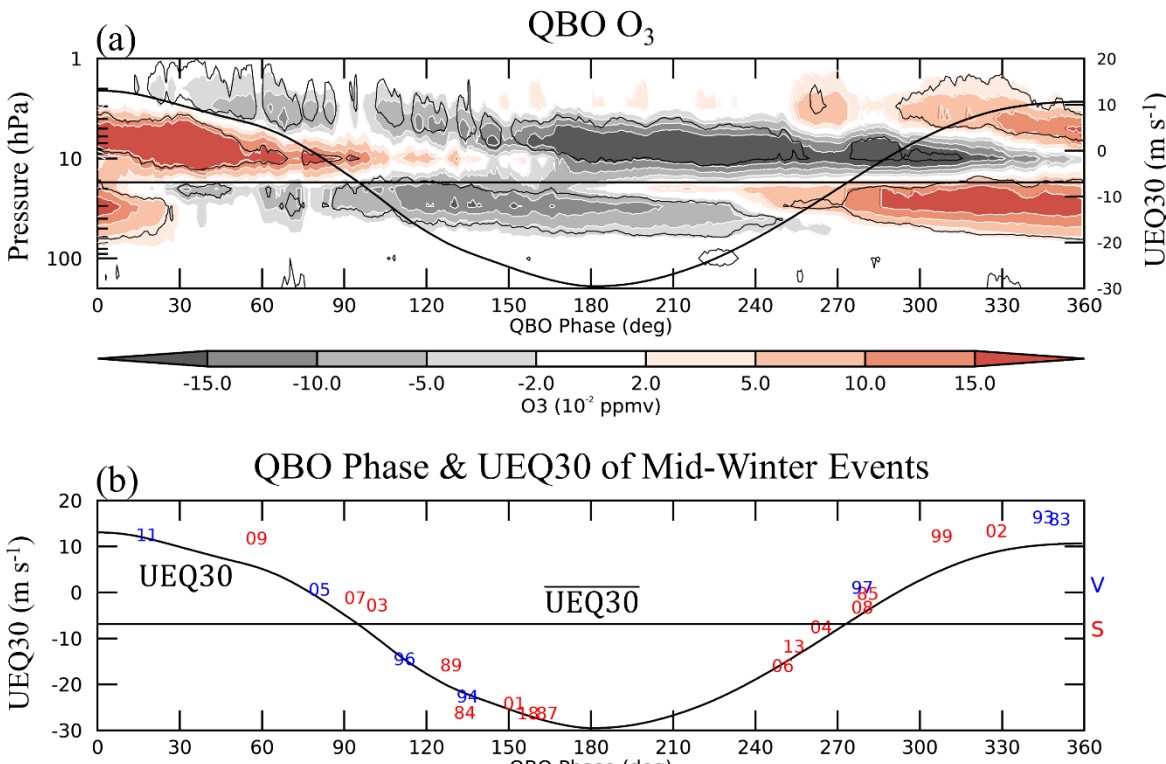

**Figure 4. Composites for QBO events. (a) QBO influences on tropical ozone; shading shows composite tropical ozone anomalies (±15°) from 16 QBO cycles (1980-2018); black contours represent statistical significance at the 95% level. A QBO cycle is defined by two consecutive positive UEQ30 maxima. (b) Central date timing of selected mid-winter stratospheric circulation events relative to the QBO phase. Red (blue) numbers indicate years and QBO phase of SSWs (VIs); S and V on the right axis is the mean UEQ30 of all SSWs and all VIs, respectively.**





**Figure 5. As Fig. 2, except for tropical ozone (±15°) and the exclusion of the eddy flux convergence term. Contours in (e-h) are the ±0.05 ppmv ozone anomalies from (c-d).**