# Peer review of "Local and Remote Response of Ozone to Arctic Stratospheric Circulation Extremes"

_Atmospheric Chemistry and Physics, 2020_

## Referee Comment (RC1) · Anonymous Referee #1 · 16 Sep 2020

The paper presents an analysis of stratospheric ozone anomalies associated with sudden stratospheric warmings (SSW). vortex intensification events (VI), as well as FW events at the end of the corresponding winters. MERRA-2 data is used and both the Arctic and the Tropics are examined. The transport mechanisms are examined using the Transformed Eulerian Mean formalism. The paper is very well written, the methods are valid and the interpretation of the results is correct. I only have minor comments that should be addressed before publication.

*Minor comments*

[Figure]

- General: I find interesting the approach of interpolating the time axis in order to get common composited times for the SSW/VI event and for the FW. It helps bring out the outstanding ozone feature during the FW following VI events. However, I am not convinced by the terminology "the event's duration" referring to the lapse time between the event's central date and the FW. The final warming terminates the winter season, and while this indeed terminates the VI events, the SSW ends when the vortex recovers, not when it breaks down.

- L134-138: I understand that S does not provide information on the photochemical changes as it is obtained as a residual and there are likely important numerical errors that prevent closing the budget, especially in a reanalysis system, where assimilation increments are included. However, you could check if the expected behavior is found in the S anomalies when referring to changes in photochemistry, e.g. for VI events in the polar region (L252-254) and in the tropics (L325-326).

- Sections 3.1 and 4.1. Several of the features described in these sections have been previously shown in the article of de la Cámara et al. 2018 JGR (https://doi.org/10.1002/2017JD028007), such as the tropical upwelling or the wave activity for SSW composites, in both reanalysis and model data.

- L157: "one-tailed t-Student test". This should be a two-tailed test, since the sample anomalies could be overestimating or underestimating the population anomalies (i.e. the null hypothesis is $\mu = 0$, not $\mu \leq 0$). This is important since for a 95% confidence level for the 8 VI events, $t_{0.025,7} = 2.365$ should be used instead of $t_{0.050,7} = 1.895$.

- L241-242: The small role of photochemical effects hypothesized here against the findings of Sagi et al. (2017) is consistent with the photochemical term shown in

de la Cámara et al. (2018) ACP. Note that this paper does show the transport and chemistry relative contributions referred to L353-355 in a CCM, and that these CCM transport results are overall consistent with your reanalysis results.

- L304: These values are much lower than their Arctic counterparts, what is the relative TOC change?

*Technical*

- L35 onward: Consider changing "transports" to "transport" throughout the paper?

- L125: change "p-coordinates" to "pressure coordinates"

- L341: "spectacular" Perhaps a more scientific term could be used (sudden/abrupt)?

---

## Referee Comment (RC2) · Anonymous Referee #2 · 22 Sep 2020

This study analyzes ozone anomalies associated with stratospheric sudden warming, vortex intensification and final warming events based on MERRA-2 reanalysis data. Long-lasting anomalies are found both in over the Arctic and in the tropics following these extreme events. In particular, the ozone anomalies only become apparent after the QBO-related signals are removed. It is a useful exercise to document the evolution and distribution of ozone anomalies following the stratospheric circulation extremes. The paper is logically organized and clearly written. I have a few comments regarding some of the methodology and results. I recommend publication of the paper after these comments are addressed.
1. Removal of the QBO signals. It is not surprising that the SSW-related ozone anomalies are masked by the QBO-related anomalies, but it is somewhat surprising that authors tried several methods to filter out QBO and only one worked. What about the linear regression with a QBO index such as in Randel and Wu (2015 JAS)?The description of the method the authors chose ("subtract the preexisting ozone anomalies of each event from its subsequent daily ozone fields") is not clear to me. The authors cited Gomez-Ecolar et al. (2014) for the method. But what described here does not seem to agree with any of the three methods described in Gomez-Ecolar et al. (2014). It sounds like calculating the difference of ozone between different periods. Then the resulting is actually ozone tendency rather than ozone anomalies itself.

2. Why there is a persistent minimum at about 20 hPa in the QBO-related ozone anomalies shown in Fig.4a? This feature seems unrealistic and is not seen in other studies (e.g. Fig. 1 of Tweedy et al. 2017 ACP).

3. The authors show the ozone tendency due to residual mean circulation and eddy flux convergence, and the eddy convergence term contribute significantly over the Arctic. Can the authors elaborate a bit more on the physical process associated with the eddy convergence, especially what determines the sign of this term?

4. Line 118: "180-day smoothed" Do the authors mean a running mean with 180 day window?

5. Line 208-209: Why is the magnitude of the ozone anomalies associated with SSW differ so much between Hocke et al. (2015) and this study?

6. Line 225: From Fig. 2, the FW anomalies following a VI event is stronger and extends to the lower stratosphere.

**ACPD**

---

## Referee Comment (RC3) · Anonymous Referee #3 · 28 Sep 2020

This manuscript shows the dynamical features of SSW and VIs from the boreal early winter to late spring in both region, Arctic and Tropics. The present study found new aspect on the dynamical impact of the final warming at the case of Vis winter. Further, there are descriptions of ozone fields at Arctic and tropics on SSW and Vis winters. The manuscript was well written, the present manuscript will be published after modifying some minor corrections and/or answer to the reviewer comments.

Minor comments:

Abstract: It is interesting points that should be described in the Abstract, that is the quantitative discussion of ozone change (Figs.2 (a,b) and Figs.5 (a,b)). The ozone

change was large at both case of SSW and Vis with FW in the Arctic, on the other hand, the ozone change was small but same amount for both case in the tropics.

Line 41 "become easterly": The major warming event accepts the reversal of zonal wind direction from westerly to easterly, however the warming event does not always reverse the wind direction, like for the minor warming.

Line 253-255 "In contrast, the negative anomalies...": Figs.2 (c, d) : Do Figs.2 (c,d) show the anomaly from climatology? If so, the authors should add the description of the anomaly from what.

Line 264 "the variations of w (Fig.3a)": If the QBO variation remains in the residual vertical velocity and temperature fields, the authors should note the fact.

Line 278 Fig4b: The year of 2016 for Vis case is absent.

---

## Author Comment (AC2) · 24 Nov 2020

The comment was uploaded in the form of a supplement:
https://acp.copernicus.org/preprints/acp-2020-790/acp-2020-790-AC2-supplement.pdf
* * *

---

## Author Comment (AC4) · 24 Nov 2020

Dear reviewers,

We also apply the following changes in the revised manuscript to better describe the work of Lubis et al. (2017):

L355-358: "We also did not explicitly consider so-called Downward planetary Wave Coupling events (DWCs) (Lubis et al., 2017), relatively short-lived events (< 10 days) associated with increases in ozone before and decreases during the event, leading to a relatively small net response. Our VI events also need to be distinguished from

so-called reflective winters, introduced by Shaw and Perlwitz (2013) and discussed by Lubis et al. (2017) to indicate winters in which wave reflection dominates. Although defined in different ways, there is some overlap between years with VIs and reflective winters and they are both associated with negative anomalies in wave driving and ozone."

---

## Author Response (AR2)

**Editor Comments**

*Comments to the Author:*
*The referees' comments on the first version of the paper were broadly favourable and it seems to me that you have responded thoroughly to their comments. Therefore I do not see it as necessary to consult the referees further. However there are some minor points that I think should be addressed before publication -- I have listed these below. Some are minor points regarding choice of words, but others are concerned with points that concerned the referees -- the 'duration' of an event and the way in which the QBO is taken into account.*

*Please can you consider these points and provide very brief responses. I then expect to be able to accept the paper.*

*l35: 'mostly controlled by transport' (not 'transports')*

Corrected.

*l37: 'seasonality of transport' (not 'not transports')*

Corrected.

*l60: 'transport'*

Corrected.

*l158, l159: You have replaced 'duration' by 'period' in response to a referee's comment, but I don't think that this addresses the referee's concern. In fact the use of 'period' adds to the confusion. I think that the key point is that defining the duration of an event as the length of time between the 'central date' and the FW is not an obvious choice of definition. Why is the end of an SSW or VI event defined by the following FW -- most readers would think of the FW as a separate/different event. I think that you simply have to say at the beginning of this section that you will characterise each event by the length of time between the central date and the FW -- and preferably give a very brief explanation of why that choice is made.*

We follow your suggestion and revise the manuscript as follows:

*L156-159: However, in the present study, we are interested in the behavior of ozone during the entire life cycle of stratospheric circulation events, beginning in December before the onset and ending with the FW at the end of winter. Our interest in this rather long period is rooted in the fact that the events and their ozone anomalies can be quite persistent, and that the FW represents yet another perturbation to the preexisting ozone fields. Since each event and FW occur at different dates, it is useful to measure the time between the central date of an event and its associated FW. This is denoted as the "length of time".*

*l281: 'local effects from events like the QBO' -- first of all it seems confusing to describe the QBO as an 'event', given that the SSW and VI events on which you focus are associated with variation on timescales of 1 week to 2 months, whereas QBO variations are on timescales of several months. The way in which the QBO might confuse interpretation of Figure 3 was if SSWs or VIs systematically occurred in certain phases of the QBO -- in which case part of what one was seeing in the composite pictures would essentially be a QBO signal.*

Yes, this might be a concern, but it turns out that the selected circulation events cover a sufficient number of random QBO phase to largely average out the QBO effect. To explain this better, we now write:

L280-281: *"Note that no filtering has been applied to this figure and that the shown changes can be due to both the remote impacts from the Arctic circulation events and the local effects from the internal variability associated with the QBO. However, the Arctic circulation events occur mostly random with respect to the QBO phase, so that the compositing largely removes possible QBO effects from the shown dynamical fields. This is also supported by the fact that Fig. 3 does not resemble the known influences of the QBO phases on the dynamics (e.g., Coy et al., 2016; their Fig. 8)."*

*l306: 'Fig. 4b demonstrates that SSWs and VIs occur during virtually any phase of the QBO, making it difficult to cleanly separate the ozone changes from the Arctic and the QBO.' -- this seems an odd statement to me -- if the SSWs and VIs occur essentially independently of QBO phase then surely that makes it easier, not more difficult to separate the VI/SSW effects?*

Agreed, this does not make sense (we don't even remember why we wrote this, it might be some leftover from a previous version of the manuscript). We now simply remove this part of the sentence:

l306: *"Fig. 4b demonstrates that SSWs and VIs occur during virtually any phase of the QBO."*

*Figure 4 caption: 'purposefully' should be 'purposely'*

Corrected.
*The paper presents an analysis of stratospheric ozone anomalies associated with sudden stratospheric warmings (SSW). vortex intensification events (VI), as well as FW events at the end of the corresponding winters. MERRA-2 data is used and both the Arctic and the Tropics are examined. The transport mechanisms are examined using the Transformed Eulerian Mean formalism. The paper is very well written, the methods are valid and the interpretation of the results is correct. I only have minor comments that should be addressed before publication.*

*Minor comments*

*• General: I find interesting the approach of interpolating the time axis in order to get common composited times for the SSW/VI event and for the FW. It helps bring out the outstanding ozone feature during the FW following VI events. However, I am not convinced by the terminology "the event's duration" referring to the lapse time between the event's central date and the FW. The final warming terminates the winter season, and while this indeed terminates the VI events, the SSW ends when the vortex recovers, not when it breaks down.*

Yes, we agree that "event duration" can be misleading. In the revised manuscript we now use "period" to better describe the time between the central date and the FW.

*• L134-138: I understand that S does not provide information on the photochemical changes as it is obtained as a residual and there are likely important numerical errors that prevent closing the budget, especially in a reanalysis system, where assimilation increments are included. However, you could check if the expected behavior is found in the S anomalies when referring to changes in photochemistry, e.g. for VI events in the polar region (L252-254) and in the tropics (L325-326).*

We performed the suggested analysis. Over the Arctic (Fig. R1) and in the upper stratosphere (above 10 hPa), the source term *S* is mostly anticorrelated with temperature *T*, as expected. Interestingly, during VIs and over the Arctic, there is some indication for chemical ozone depletion, consistent with previous studies (Isaksen et al., 2012; Manney et al., 2011, 2020). However, *S* over the Tropics (Fig. R2) does not follow this pattern and largely opposes the tendencies due to the vertical advection (Fig. 5g and 5h). We now mention this in the manuscript at line 255:

L255: *"We also examined the source term S (not shown) and find negative tendencies in the lower stratosphere (10 – 100 hPa) during and after the onset of VIs, indicative for temperature-driven heterogeneous ozone depletion as suggested by previous studies (Isaksen et al., 2012; Manney et al., 2011, 2020). In the upper stratosphere, S is as expected mostly anticorrelated with T."*

**Reference**
Manney, G. L., Livesey, N. J., Santee, M. L., Froidevaux, L., Lambert, A., Lawrence, Z. D., Millán, L. F., Neu, J. L., Read, W. G., Schwartz, M. J., Fuller, R. A.: Record-low Arctic stratospheric ozone in 2020: MLS observations of chemical processes and comparisons with previous extreme winters, Geophys. Res. Lett., 47, e2020GL089063, https://doi.org/10.1029/2020GL089063.

[Figure]

**Figure R1. Composite anomalies for (left) SSWs and (right) VIs over the Arctic. Shown are time-height cross-sections for (a-b) temperature (K) (65°N-90°N) and (c-d) ozone tendency associated with *S* (ppbv day⁻¹).**

[Figure]

**Figure R2. As Fig. R1, except for the Tropics (±15°).**

• *Sections 3.1 and 4.1. Several of the features described in these sections have been previously shown in the article of de la Cámara et al. 2018 JGR (https://doi. org/10.1002/2017JD028007), such as the tropical upwelling or the wave activity for SSW composites, in both reanalysis and model data.*

We thank the reviewer for pointing this out. We now reference the paper to support its results.

**• L157: "one-tailed t-Student test". This should be a two-tailed test, since the sample anomalies could be overestimating or underestimating the population anomalies (i.e. the null hypothesis is $\mu = 0$, not $\mu \leq 0$). This is important since for a 95% confidence level for the 8 VI events, $t_{0.025,7} = 2.365$ should be used instead of $t_{0.050,7} = 1.895$.**

We agree with the reviewer. In the revised manuscript, we now use for the figures a two-tailed t-test at the 95% confidence level. Note that our results suggest that the anomalies of temperature and ozone during VIs are still significant after the change.

**• L241-242: The small role of photochemical effects hypothesized here against the findings of Sagi et al. (2017) is consistent with the photochemical term shown in de la Cámara et al. (2018) ACP. Note that this paper does show the transport and chemistry relative contributions referred to L353-355 in a CCM, and that these CCM transport results are overall consistent with your reanalysis results.**

We also compared our ozone tendency results with de la Cámara et al. (2018). Although we use a different vertical system, the two results are quite consistent with each other in most of the stratosphere for SSW events. We followed your advice and changed our paper in two locations:

L241: *"Overall, this indicates that the decrease in mid-stratospheric ozone after SSWs is mainly of dynamical origin, consistent with de la Cámara et al. (2018). We note that this does not support the ideas of Sagi et al. (2017), who argue that the ozone decrease is due to chemical reactions involving NOx species."*

L353: *"Nevertheless, it would be interesting to evaluate the relative contributions from the dynamics and the chemistry in changing ozone during SSWs and VIs, using output from a range of coupled chemistry climate models (CCMs), similar in spirit to de la Cámara (2018) for SSWs using the WACCM model."*

**• L304: These values are much lower than their Arctic counterparts, what is the relative TOC change?**

Fig. R5 shows the SSW and VI composites for column ozone as in Fig. 2 and Fig. 5. The blue lines in Fig. R5 are the percent column ozone anomaly with respect to climatology. The result suggests that changes in column ozone amount to 10%-12% of climatological values over the Arctic, while the ozone anomaly over the Tropics amount to only 0.5%-1%.

[Figure]

**Figure R5. Column ozone composites (left) SSWs and (right) VIs. Blue line (right axis) represents the percentage of column ozone anomaly with respect to climatology.**

We now add this information to our paper by adding/modifying the following sentences:

L301: *"… during SSWs and VIs. The variations in tropical column ozone are rather small and amount to only ~0.5 – 1% of the climatological values, which can be compared to the 10 – 15% changes seen over the Arctic. Nevertheless, the changes in tropical ozone are quite coherent and persistent. For SSWs, …"*

L303-305: *"SSWs are followed by a small reduction in tropical column ozone by ~2.5 DU (~ -1 %) and an increase by ~1-2 DU (~ 0.5 %) after mid-March, which persists until late spring."*

L310: *"During VIs (Fig. 5b), there are small tropical column ozone anomalies, which are mostly positive (~1 DU or 0.5%) and only become negative (~2 DU or 1%) after the FW."*

In the revised manuscript, we also replaced the column ozone tendency line (green lines in Fig. 2a-2b and Fig. 5a-5b) with the percentage of ozone anomalies (as in Fig. R5).

***Technical***

• **L35 onward: Consider changing "transports" to "transport" throughout the paper?**

This has been corrected.

• **L125: change "p-coordinates" to "pressure coordinates"**

This has been corrected.

• **L341: "spectacular" Perhaps a more scientific term could be used (sudden/abrupt)?**

We now use the word "remarkable" instead.
*This study analyzes ozone anomalies associated with stratospheric sudden warming, vortex intensification and final warming events based on MERRA-2 reanalysis data. Long-lasting anomalies are found both in over the Arctic and in the tropics following these extreme events. In particular, the ozone anomalies only become apparent after the QBO-related signals are removed. It is a useful exercise to document the evolution and distribution of ozone anomalies following the stratospheric circulation extremes. The paper is logically organized and clearly written. I have a few comments regarding some of the methodology and results. I recommend publication of the paper after these comments are addressed.*

*1. Removal of the QBO signals. It is not surprising that the SSW-related ozone anomalies are masked by the QBO-related anomalies, but it is somewhat surprising that authors tried several methods to filter out QBO and only one worked. What about the linear regression with a QBO index such as in Randel and Wu (2015 JAS)? The description of the method the authors chose ("subtract the preexisting ozone anomalies of each event from its subsequent daily ozone fields") is not clear to me. The authors cited Gomez-Ecolar et al. (2014) for the method. But what described here does not seem to agree with any of the three methods described in Gomez-Ecolar et al. (2014). It sounds like calculating the difference of ozone between different periods. Then the resulting is actually ozone tendency rather than ozone anomalies itself.*

We did not use linear regression in our analysis. Instead, we tested a similar method, taking into account the mean QBO ozone structure (Fig. 4a) and QBO phase during each event. However, we were not satisfied with the result and therefore used another filtering method.

Our filtering method defines a pre-existing QBO ozone signal from the mean ozone anomalies over day -60 to day -30 with respect to the SSW/VI central date. We subtract this QBO signal from the ozone anomalies associated with each circulation event before taking composites. This method assumes that the QBO time scale is much longer than the time scale of SSWs or VIs. Our method is similar to Gómez-Escolar et al. (2014) (Fig. 7 in their study) and Kodera (2006), however we defined a somewhat different time period for removing the QBO signal than the previous two studies.

To make this clearer, we reworded our manuscript as follows:

L291-299: *"...by Dunkerton et al. (1988). To filter out the QBO influences from the tropical ozone, we define the QBO ozone signal as the mean ozone anomalies over day -60 to day -30 with respect to the SSW/VI central date, which is then subtracted*

*from the ozone associated with each Arctic circulation event. We use the resulting ozone anomalies for preparing Figs. 5c and 5d."*

**2. Why there is a persistent minimum at about 20 hPa in the QBO-related ozone anomalies shown in Fig.4a? This feature seems unrealistic and is not seen in other studies (e.g. Fig. 1 of Tweedy et al. 2017 ACP).**

Our result of the QBO ozone composite (Fig. 4a) is consistent with Tweedy et al. (2017) (Fig. R6), which only shows ozone anomalies between 10 and 70 hPa (right axis of their figure). The QBO ozone composite of Tweedy et al. (2017) (Fig. R6a) reveals a nodal point for ozone minimum between 10 and 20 hPa that is also seen in our result of Fig. 4a. As requested, we show below in Fig. R7 (black line) the evolution of ozone at 20 hPa, suggesting that the 20 hPa ozone undergoes a sign change within the QBO cycle. Another reason for the apparent discrepancies is that Tweedy et al. show the anomalies in percent, whereas we show them as absolute anomalies (in ppmv).

To clarify the similarity between Tweedy et al. (2017) and our result, we now reword our manuscript as follows:

L283: *"Our results (Fig. 4a) are in good agreement with their study, e.g., there is a nodal point of small ozone variations between 10 and 20 hPa, with much stronger variations above and below."*

[Figure]

**Figure R6. Time-height cross-sections for QBO ozone. Shown are ozone anomalies for (a) QBO composite, (b) 2015-2016 QBO event and (c) b-a. Adapted from Tweedy et al. (2017), Fig. 1.**

[Figure]

**Figure R7. Composite tropical ozone anomalies for QBO events. Shown are anomalies at (blue) 10 hPa, (black) 20 hPa, and (red) 30 hPa.**

**3. The authors show the ozone tendency due to residual mean circulation and eddy flux convergence, and the eddy convergence term contribute significantly over the Arctic. Can the authors elaborate a bit more on the physical process associated with the eddy convergence, especially what determines the sign of this term?**

To answer your question, we revised our manuscript and explained in more detail the meaning of the eddy flux convergence term in Sect. 2.3:

L132: *"The eddy flux convergence contains effects that are not explained by the advection of zonal mean ozone by the zonal mean circulation. The convergence is associated with transports of zonal disturbances in ozone by zonal disturbances in meridional or vertical velocity. In the stratosphere, these disturbances (or eddies) are primarily due to upward propagating planetary waves. The convergence term indicates that covariance between eddy velocities and ozone can transport ozone, and that where this eddy ozone flux converges a zonal mean ozone tendency can be induced. For example, a northward ozone flux is created if the signs of the meridional velocity and the ozone perturbations tend to be the same, and if this flux decreases in the northward direction (converges), it would create a positive ozone tendency in the zonal mean. Our result (not shown) suggests that the meridional component of the eddy flux convergence (the first term of the $\boldsymbol{M}$-vector in equation (1) dominates the vertical component over most of the stratosphere."*

**4. Line 118: "180-day smoothed" Do the authors mean a running mean with 180 day window?**

Yes, we understand our wording is not clear. We now revise our manuscript as follows:

L118: *"We use a 180-day running mean window to smooth the zonal-mean equatorial (±5°) zonal wind at 30 hPa (UEQ30) and determine the phase of the QBO."*

**5. Line 208-209: Why is the magnitude of the ozone anomalies associated with SSW differ so much between Hocke et al. (2015) and this study?**

The differences are only apparent. While we show area-weighted latitudinal averages over 65°N-90°N (with a maximum of ~50 DU), Fig. 2 in Hocke et al. (2015) shows larger ozone anomalies (up to 90 DU) only close to the pole (which represents a small area). Therefore, we believe that our result is consistent with Hocke et al. (2015).

In the revised manuscript, we add a clarification at line 209 as follows:

L209: *"However, we note that the differences are only apparent, as we show area-weighted latitudinal averages of column ozone and as the extreme ozone increases in Hocke et al. (2015) occur only close to the pole."*

**6. Line 225: From Fig. 2, the FW anomalies following a VI event is stronger and extends to the lower stratosphere.**

Yes, actually there are some negative ozone anomalies in the lower stratosphere at the FW following the VIs. We revised our manuscript as follows:

L224-226: *"The structure of these anomalies is somewhat similar to that of SSWs, except that they are weakly negative in the lowermost stratosphere."*
*This manuscript shows the dynamical features of SSW and VIs from the boreal early winter to late spring in both region, Arctic and Tropics. The present study found new aspect on the dynamical impact of the final warming at the case of Vis winter. Further, there are descriptions of ozone fields at Arctic and tropics on SSW and Vis winters. The manuscript was well written, the present manuscript will be published after modifying some minor corrections and/or answer to the reviewer comments.*

*Minor comments:*

*Abstract: It is interesting points that should be described in the Abstract, that is the quantitative discussion of ozone change (Figs.2 (a,b) and Figs.5 (a,b)). The ozone change was large at both case of SSW and Vis with FW in the Arctic, on the other hand, the ozone change was small but same amount for both case in the tropics.*

We revised the abstract by adding specific numbers for the changes in column ozone in the following way:

L15-17: *"Over the Arctic and during sudden warmings, ozone undergoes a rapid and long-lasting increase of up to ~50 DU, which only gradually…. In contrast, vortex intensifications are passive events, associated with gradual decreases in Arctic ozone that reach ~40 DU during late winter and decay thereafter."*

L20-21: *"After controlling for this effect, small but coherent reductions in tropical ozone can be seen during the onset of sudden warmings (~2.5 DU), and also during the final warmings that follow vortex intensifications (~2 DU)."*

*Line 41 "become easterly": The major warming event accepts the reversal of zonal wind direction from westerly to easterly, however the warming event does not always reverse the wind direction, like for the minor warming.*

We now revised the manuscript as follows:

L37-38: *"At times, the bursts of waves and their interaction with the polar vortex are strong enough to create so-called major Stratospheric Sudden Warming Events (SSWs)…"*

*Line 253-255 "In contrast, the negative anomalies. . .": Figs.2 (c, d) : Do Figs.2 (c,d) show the anomaly from climatology? If so, the authors should add the description of the anomaly from what.*

Yes, Figs. 2c and 2d are daily ozone anomalies, and we corrected the caption of figure 2 to makes this clearer:

Caption 2: *"Figure 2. Arctic ozone …. Remaining panels are anomalous time-height cross-sections of …"*.

The calculation of daily ozone anomaly is already described in section 2.1, where we write:

L96-99: *"We compute daily climatologies from MERRA-2 by averaging each day of the year over the entire record and smoothing over the seasonal cycle using 10-day running means. Daily anomalies are obtained by subtracting the climatologies from the daily data."*

***Line 264 "the variations of w (Fig.3a)": If the QBO variation remains in the residual vertical velocity and temperature fields, the authors should note the fact.***

Yes, the QBO variations are still contained in the diagnostic of the dynamical quantities. To make this clearer, we now add one sentence after line 263:

*"Note that no filtering has been applied to this figure and that the shown changes are due to both the remote impacts from the Arctic circulation events and the local effects from events like the QBO."*

***Line 278 Fig4b: The year of 2016 for Vis case is absent.***

As described in section 2.2, we did not include the anomalous 2015-2016 QBO event in our analysis. We revised the caption of Fig. 4 as follows:

*Caption 4: "Figure 4. Composites for QBO events …; S and V on the right axis is the mean UEQ30 of all SSWs and all VIs (except 2016), respectively. The 2015-2016 QBO event has been purposefully excluded from this analysis due to the anomalous nature of this event.*
We also apply the following changes in the revised manuscript to better describe the work of Lubis et al. (2017):

[revised manuscript text omitted]